# Explainable Artificial Intelligence (XAI) in Pain Research: Understanding the Role of Electrodermal Activity for Automated Pain Recognition

**DOI:** 10.3390/s23041959

**Published:** 2023-02-09

**Authors:** Philip Gouverneur, Frédéric Li, Kimiaki Shirahama, Luisa Luebke, Wacław M. Adamczyk, Tibor M. Szikszay, Kerstin Luedtke, Marcin Grzegorzek

**Affiliations:** 1Institute of Medical Informatics, University of Lübeck, Ratzeburger Allee 160, 23562 Lübeck, Germany; 2Faculty of Informatics, Kindai University, Higashiosaka 577-8502, Osaka, Japan; 3Department of Physiotherapy, Pain and Exercise Research Luebeck (P.E.R.L.), Institute of Health Sciences, University of Lübeck, Ratzeburger Allee 160, 23562 Lübeck, Germany; 4Laboratory of Pain Research, Institute of Physiotherapy and Health Sciences, The Jerzy Kukuczka Academy of Physical Education, 40-065 Katowice, Poland; 5Department of Knowledge Engineering, University of Economics in Katowice, Bogucicka 3, 40-287 Katowice, Poland

**Keywords:** pain recognition, machine learning, deep learning, hand-crafted features, physiological signals, pain perception, explainable artificial intelligence

## Abstract

Artificial intelligence and especially deep learning methods have achieved outstanding results for various applications in the past few years. Pain recognition is one of them, as various models have been proposed to replace the previous gold standard with an automated and objective assessment. While the accuracy of such models could be increased incrementally, the understandability and transparency of these systems have not been the main focus of the research community thus far. Thus, in this work, several outcomes and insights of explainable artificial intelligence applied to the electrodermal activity sensor data of the PainMonit and BioVid Heat Pain Database are presented. For this purpose, the importance of hand-crafted features is evaluated using recursive feature elimination based on impurity scores in Random Forest (RF) models. Additionally, Gradient-weighted class activation mapping is applied to highlight the most impactful features learned by deep learning models. Our studies highlight the following insights: (1) Very simple hand-crafted features can yield comparative performances to deep learning models for pain recognition, especially when properly selected with recursive feature elimination. Thus, the use of complex neural networks should be questioned in pain recognition, especially considering their computational costs; and (2) both traditional feature engineering and deep feature learning approaches rely on simple characteristics of the input time-series data to make their decision in the context of automated pain recognition.

## 1. Introduction

Artificial intelligence and particularly deep learning methods have achieved outstanding results for various Machine Learning (ML) tasks in the past few years. In particular, modern Deep Learning (DL) architectures yield results that are equivalent to human performance for image classification and sometimes even outperform them. It is therefore not surprising that attempts were made to develop ML models that compete with experts in their respective fields. For example, in the area of medicine, automated systems have been built to classify skin cancer [1], segment Optical Coherence Tomography (OCT) images [2], classify COVID-19 patients from chest CT images [3], classify Parkinson’s disease based on audio files [4] and many more to replace or enhance their current gold standard based on expert knowledge. Moreover, the increasing simplicity to acquire physiological sensor data by using cheap and simple-to-use wearables has opened completely new research areas such as automated emotion recognition [5], sleep stage classification [6], Human Activity Recognition (HAR) [7], hunger detection [8], and countless others. Similarly, efforts were undertaken to automatize the clinical recognition of pain. As pain constitutes both a symptom and disease [9], it is a crucial indicator in any medical application. Currently, the gold standard for pain detection is represented by Numerical Rating Scales (NRSs), which ask patients about the pain they felt using a numerical scale between 0 and 10 and its corresponding anchors “no pain” and “worst imaginable pain”. While there are several versions of this concept, such as Visual Analogue Scales (VASs) or special types of scales targeted for children (Faces Pain Scale) [10], and even pain observation tools used for patients with disorders of consciousness such as the Nociception Coma Scale (NCS) [11], these questionnaires have several drawbacks. On the one hand, pain is a highly subjective concept that is hard to communicate and is influenced by past experiences. On the other hand, even accurate estimates of pain identified by the gold standard are sporadic rather than continuous measurements. Further, these questionnaires fail whenever patients are unable to (reliably) communicate their pain, such as with coma patients, elderly (dementia) patients or children. Thus, various medical applications would benefit from a continuous, objective and automated pain recognition system. In the past, several attempts to build such systems were made by training a ML model to associate physiological signals with their corresponding pain categories. Initial efforts relied on the use of classical ML models trained on features obtained from expert knowledge, such as Support Vector Machines (SVMs) [12]. More recent publications evaluated the use of DL approaches to overcome the need for explicit expertise in pain physiology [13]. Figure 1 summarizes the current trend of work in the area of automated pain assessment. It visualizes the number of published papers for each year since 2000 that were found through different search engines. A search using the website “dimensions.ai” in combination with the search keys “machine learning pain” and “deep learning pain” in the title and abstract was conducted. In addition, a PubMed search was performed with “pain machine learning[title/abstract]” as the search term. Furthermore, a Google Scholar retrieval for “machine learning pain” and “deep learning pain” was performed (searching for papers having all keywords in the title). The results show an increasing amount of publications over time, which reflects the rise of interest and importance of the research field in recent years.

The creation of publicly available datasets, such as the UNBC-McMaster shoulder pain expression archive database [14] and BioVid Heat Pain Database (BVDB) [15], significantly increased the number of new ML models in the scope of automated pain recognition. Cited numerous times, the BVDB induced pain-inducing heat stimuli with varying intensity. While applying four calibrated heat temperatures (called $T_{1},\ldots ,T_{4}$) corresponding each to a different pain level, the physiological signals Electromyogram (EMG), Electrocardiogram (ECG) and Electrodermal Activity (EDA) were recorded. Usually, pain recognition tasks are then built in a binary way, distinguishing between baseline data associated with no pain ($T_{0}$) and data associated with a certain pain level. Because it consistently led to the best classification results, the task $T_{0}$ vs. $T_{4}$, i.e., “no pain” vs. “high pain”, was investigated intensively in the past. Table 1 summarizes previously published results achieved on the BVDB. A more detailed description of the dataset can be found in Section 2.1.2.

Previous outcomes on the automated recognition of pain can be summarised as follows. The classification of physiological sensor data yields better results compared to the ones based on behaviour input such as video data [21,22,23,24,25,26,27]. Regarding physiological modalities, EDA was detected as the individual modality with the highest impact on the classification outcome [13,20,26,28]. In addition, feature engineering and learning perform roughly equivalently, where tasks such as “no pain” vs. “low pain” remain relatively challenging in contrast to “no pain” vs. “high pain” [20]. In contrast, other work has shown that feature learning can outperform approaches based on Hand-Crafted Features (HCFs) [13]. Finally, the use of a subjective pain label (patient feedback) can boost the classification performance of such systems [20,29]. The performance of the presented systems and accuracy of such ML models were increased incrementally over time. However, despite the numerous works published in this area each year, the understandability and transparency of these systems were not thoroughly investigated by the research community. However, the medical field could benefit from insights created by interpretable ML models that would help them grasp a deeper understanding of pain. Moreover, knowing what ML models rely on to make their decision could be investigated to further fine-tune them. Thus, in this work, several ML models are evaluated and compared on two different datasets for automated pain recognition. First, classical ML models such as Random Forest (RF) based on HCFs are trained and analysed with Recursive Feature Elimination (RFE) to determine the most important data characteristics. Then, interpretability approaches such as Gradient-weighted Class Activation Mapping (Grad-CAM) are similarly applied to DL models such as Convolutional Neural Networks (CNNs). Being the most discriminative modality for pain recognition, only the EDA signal is analysed in the current work. Furthermore, several outcomes and insights of Explainable Artificial Intelligence techniques for automated pain recognition are presented to further understand the mechanisms of pain in detail. The main contributions of this study are highlighted below:A comparison of various ML models based on feature engineering and end-to-end feature learning including recent state-of-the-art DL methods evaluated on the PainMonit Database (PMDB) and BVDB.The interpretation of the decisions of both HCFs and DL models in the scope of automated pain recognition using Explainable Artificial Intelligence (XAI).The proposal of rules based on simplistic manually-crafted features to distinguish between “no pain” and “high pain” using EDA data only.

Moreover, our studies highlight the following insights: (1) single simplistic features can still compete with complex DL models with millions of parameters. (2) Both approaches, based on HCFs and DL features, focus on straightforward characteristics of the given time series data in the context of automated pain recognition. The remainder of the work is organised as follows. The used datasets, models and approaches for XAI are explained in Section 2. The resulting outcomes are presented in Section 3 and discussed in Section 4. Eventually, the main conclusions are summarised in Section 5.

## 2. Materials and Methods

Two types of approaches, one based on feature engineering and the other on deep feature learning, were implemented to classify the EDA samples of two pain datasets for automated pain recognition. The XAI methods for understanding the decisions of the classifiers were realised separately for both approaches. In this chapter, Section 2.1 describes the utilised data, followed by Section 2.2, which presents the feature engineering algorithms, and Section 2.3 summarises the leveraged DL techniques.

### 2.1. Data

Previous publications evaluated various sensor modalities for the automated classification of pain. It was shown that especially EDA data are helpful for the given classification task. This section describes the analysed data in the context of automated pain recognition. Initially, the fundamentals of EDA recordings are summarised in Section 2.1.1, followed by Section 2.1.2 and Section 2.1.3 introducing the BVDB and PMDB datasets, respectively.

#### 2.1.1. Electrodermal Activity

EDA, sometimes also referred to as Galvanic Skin Response (GSR), is a measurement of small fluctuations in the conductance of the skin. It is measured by applying a low and undetectable voltage to the skin and monitoring the changes in conductance afterwards. While in the past, larger devices in combination with (wet) electrodes were used to measure it, various wearable devices such as wristbands are able to detect EDA today as well. Usually, EDA measurements are given in micro Siemens (μS) and present values greater than 0 and smaller than 20. The readings are directly related to the sweat secretion on the electrode site, which is linked to the Autonomic Nervous System (ANS), and therefore changes unconsciously and cannot be controlled voluntarily. Here, an increase in the arousal of the sympathetic branch of the ANS leads to increased sweat gland activity, which is visible by a rising EDA. Currently, EDA measurements are leveraged for various ML tasks. Correlated with psychological or physiological arousal, it is often used in emotion recognition [30]. Several publications proved that automated pain recognition models can be trained using EDA data [13,20,29]. In addition, it was shown that even small differences in the applied pain stimulus lead to changes in the EDA curves [31]. The analysis of EDA discriminates between the slowly changing Skin Conductance Level (SCL), also referred to as tonic level, and the smaller spikes in the signal called Skin Conductance Responses (SCRs), sometimes also referred to as phasic information. These SCRs can be event-related and triggered by external stimuli, based on motor activity or just spontaneously occur without an impulse or event [32]. Figure 2 shows an EDA recording during a painful heat stimulus.

#### 2.1.2. BioVid Heat Pain Database

Published in 2013 by Walter et al. [15], the BioVid Heat Pain Database (BVDB) is one of the first publicly available datasets for automated pain recognition. Pain was induced via a thermode (Medoc, Ramat Yishai, Israel) at the right arm of participants while capturing physiological sensors and video recordings. In total, the data from 90 subjects were recorded. Unfortunately, 3 subjects were excluded because of technical issues during the data acquisition. To take into account the subjective perception of pain, individual temperature stimuli were found during a calibration phase first. Initially, the temperatures when the heat becomes painful (TP) and unbearable, also referred to as pain tolerance (TT), were estimated by slowly increasing the thermode starting at 32 °C. The found thresholds were used to define 4 pain intensities by using temperatures referred to as $T_{1}$ to $T_{4}$ that are evenly distributed between TP and TT. Moreover, the data of the baseline temperature (32 °C) were considered non-painful ($T_{0}$), resulting in 5 distinguished classes. Twenty repetitions of each stimulus were applied for 4 s with 8–12 s of randomised pauses in between for two different sensor setups and pain induction phases each. Two data subsets, referred to as Part A and Part B, were acquired. Part A includes video recordings of the subjects’ faces as well as 3 physiological sensor modalities described below. Part B acquired the same data, but replaced the video stream with facial EMG. As Part A is the more used and cited one, it is used in this study as well and referred to as BioVid Heat Pain Database (BVDB) from now on, for simplicity reasons. The following sensor modalities were recorded by a Nexus-32 amplifier:Electrodermal Activity (EDA): The EDA, also referred to as GSR, was measured between the index and ring finger.Electrocardiogram (ECG): The participants’ heart rate activity was recorded using two electrodes, one on the upper right and one on the lower left of the body.Electromyogram (EMG): Using a two-channel surface electromyogram (sEMG), the activity of the shoulder muscle (Trapezius) was recorded.

The time series data of the different modalities were resampled to a common frequency of 512 Hz. The samples of the finalised dataset consist of 5.5 s windows with a delay of 3 s after the onset stimulus. The resulting data format for the dataset is (subjects × stimuli × repetitions, sampling rate × windows length, sensors) = (87 × 5 × 20, 512 × 5.5, 3) = (8700, 2816, 3). Further insights about the dataset and a detailed description can be found in [15]. Hereinafter, the binary classification task $T_{0}$ vs. $T_{4}$ is used to refer to the task “no pain” vs. “high pain” in the scope of the BVDB.

#### 2.1.3. PainMonit Database

Similarly, the PainMonit Database (PMDB) [20] was acquired by inducing heat-induced pain in subjects using a Pathway CHEPS (Contact Heat-Evoked Potential Stimulator thermode, Medoc, Ramat Yishay, Israel) at the Institute of Medical Informatics, University of Lübeck, Germany. In contrast to the BVDB, several adjustments were made to the data acquisition protocol. Calibration was performed to reduce variability in pain ratings [33] by using the “method of staircase” [34] (p. 400) and performed twice to ensure further robustness by averaging the results of the two trials. In addition, the duration of the stimuli windows was raised to 10 s to ensure enough recording time of the EDA signal, as previous studies showed that peak values normally occur “between 3 and 6 s poststimulus” [35], and thus longer recording times than the ones chosen in BVDB could be beneficial. A non-painful stimulus was also added to analyse the role of external stimuli on physiological body reactions, and the total number of stimuli was reduced. Moreover, the participants were asked to rate their subjective pain in real-time with the help of a Computerised Visual Analogue Scale (CoVAS) during the induction phase to incorporate a subjective label in the dataset. Two wearable devices, the Empatica E4 (E4) (Empatica E4, Empatica Inc., Boston, MA, USA) and respiBAN Professional (RB) (respiBAN Professional, Plux, Lisbon, Portugal), registered several physiological modalities that were resampled to a common sampling rate of 250 Hz and are listed as follows:respiBAN Professional:Electrodermal Activity (EDA): The EDA was captured between the medial phalanx of the index and middle finger of the non-dominant arm.Electrocardiogram (ECG): Using 3 electrodes, one on the upper right, one on the upper left and one on the lower left of the body, heart rate activity was recorded.Electromyogram (EMG): Using a two-channel surface electromyogram (sEMG), the activity of the shoulder muscle (Trapezius) was recorded.Respiration: The breathing of the subject is recorded using a chest belt of the RB device.Empatica E4:5.Electrodermal Activity (EDA): The wristband is measuring the EDA using two electrodes inside of its strap.6.Blood Volume Pulse (BVP): Emitting green and red light and detecting the reflection using a photodiode, the Empatica is capable of estimating the BVP of its wearer.7.Inter-Beats-Interval (IBI): The Empatica calculates the time between consecutive heartbeats based on the recorded BVP information.8.Heart Rate (HR): The Empatica calculates a HR signal based on the recorded BVP information.9.Skin temperature: The Empatica reads the peripheral skin temperature.

The resulting dataset of 52 subjects has a data shape of (subjects × stimuli × repetitions, sampling rate × windows length, sensors) = (52 × 6 × 8, 250 × 10, 9) = (2496, 2500, 9). More details and an in-depth description of the dataset are introduced in [20]. Hereafter, task *B* vs. $P_{4}$ refers to the classification task of “no pain” vs. “high pain” in the scope of the PMDB. Following our previous findings comparing the performances of various sensor modalities [20], the EDA signal originating from the RB is chosen over the one provided by the E4. The PMDB is not publicly accessible due to privacy concerns.

### 2.2. Classification Based on Feature Engineering

To train traditional ML models, key characteristics are extracted from the raw data using expert knowledge. These so-called HCFs can include anything from simple statistical values to complex features achieved by tailored algorithms for a specific use case. Based on these key characteristics, ML models such as RFs are trained to learn an association between data samples and labels. Figure 3 summarises the sequence of methods applied to the pain datasets to evaluate feature engineering techniques for automated pain classification and to gain additional knowledge about the decision-making process.

#### 2.2.1. Feature Extraction

Features to automatically classify pain were calculated following [20]. Several statistical features derived from the EDA signal, such as rapid changing spikes, also called Skin Conductance Response (SCR), with its key characteristic and slowly adapting SCL, were calculated. Moreover, more complex methods, such as the derivative of phasic component of EDA (dPhEDA) based on a convex EDA optimisation method (cvxEDA) [36] and spectral features time-varying index of sympathetic activity (TVSymp), and its modified version (modified spectral features time-varying index of sympathetic activity (MTVSymp)) [37,38], were applied to retrieve more detailed characteristics. The features were extracted directly from the raw data samples of both datasets. A description of the HCFs used in this study is summarised in Table 2.

#### 2.2.2. Random Forest

RFs, first introduced by Breiman [39], are simple but effective tools that have been applied to various tasks for ML. The approach based on Bootstrap aggregating (Bagging) leverages the union of several Decision Trees (DTs) trained on varying subsets of the initial training data (bootstrapping). A classification outcome of the ensemble technique is retrieved by majority voting for classification or averaging the results of the distinct classifiers for regression (aggregation). RFs were implemented using Sklearn v1.1.3 in combination with 100 DTs, with the samples required to split an internal node set to 2, and no constraint on the maximum depth of the individual trees. A small study showing the influence of each RF hyper-parameter on the classification accuracies can be found in Section A.1. Moreover, RFs were trained for regression tasks with two outputs, one for each class in the binary configurations. The output node (class) with the higher prediction is then picked as the classification output. The impurity of the implemented RF regressors was computed as a mean squared error as described in Equation (Equation 1).

#### 2.2.3. XAI in RFs

DTs, and thus RFs, have the advantage to illustrate their decision process in the way they are structured. Each node of the classifier represents a rule on a feature to split the dataset, and thus provides information about the impact of individual features. Here, an impurity $C_{j}$ at each node *j* of a DT can be calculated as:(1)$$C_{j}=\frac{1}{N}\sum_{i=1}^{N}{\left(y_{i}-\mu \right)}^{2},\;$$
where $y_{i}$ is the label for sample *i*, *N* is the total number of samples, and $\mu =\frac{1}{N}\sum_{i=1}^{N}y_{i}$. Using this, node importance $n_{j}$ can be calculated as follows:(2)$$n_{j}=w_{j}C_{j}-w_{j}^{left}C_{j}^{left}-w_{j}^{right}C_{j}^{right},\;$$
where $w_{j}$ is the weighted number of samples reaching node *j*, and left and right annotate the left and right child nodes, respectively. The importance for every feature $f_{i}$ in the feature space can then be estimated by:(3)$$f_{i}=\frac{\sum_{j\in I_{i}}n_{j}}{\sum_{k\in N}n_{k}},\;$$
where $I_{i}$ is the set of all nodes that split on feature *i*. Usually, these importance values are normalised afterwards:(4)$$f_{i}^{norm}=\frac{f_{i}}{\sum_{j\in F}f_{j}},$$
where *F* is the set of all features. To estimate the importance of a feature in a RF, the importance associated with the feature in all trees is summed up and divided by the number of trees. These retrieved scores can also be averaged for several outcomes of a Cross Validation (CV) to estimate feature importance in this way. Moreover, these metrics can be used to implement a RFE [40] strategy. First, a RF model is trained on the complete feature space. Next, the individual features are ranked according to their importance given by their impurity score. Afterwards, the least important feature is discarded and the model is trained again. The mentioned steps are repeated several times to find an optimal number of features for the given task. Once the dispensable and redundant features for the classification tasks are removed, an optimal feature space can eventually be found.

### 2.3. Classification based on Feature Learning

In addition to the HCF approach, several DL architectures either commonly used or taken from the related literature were implemented. To start with, Multi-Layer Perceptron (MLP), CNN and Convolutional Autoencoder (CAE) models were implemented, as previous studies showed promising results for time-series classification [20]. Moreover, deep learning-based approaches that have gained a lot of traction in the past few years, such as Contrastive Learning (CL) [41] and transformer networks [42], were evaluated. As CNNs proved to be working especially well on image datasets in the past, a method to transform the time series data to image representations using Gramian Angular Fields (GAF) [43] and classify them using a CNN was tested as well. In contrast, Recurrent Neural Networks (RNNs) such as Long Short-Term Memorys (LSTMs), which are challenging and time-consuming to train and did not provide promising results in previous studies, were not considered in the current work. The complete list of used DL models is as follows:MLP;CNN;CAE;A supervised Contrastive Learning (CL) [41] architecture based on the encoder of the CAE model;A transformer network [42];Gramian Angular Fields (GAFs) [43] representations of the 1D time series data in combination with multi-dilated kernel (MDK) residual modules [44].

All methods were evaluated on both datasets. The architectures and hyper-parameters for the MLP, CNN and CAE are inspired by Gouverneur et al. [20]. The performance of the CNN was further improved by adding another block of convolution and pooling layers and increasing the number of feature maps. The CL, transformer and MDK models are implemented following the suggestion found in the source papers listed above. The basic concepts of each model are described below, and a detailed summary of the DL architectures used can be found in Section A.2. Figure 4 summarises the sequence of methods applied to the pain datasets to evaluate several feature learning techniques for automated pain classification and to gain additional knowledge about the decision-making process.

In the following sections, descriptions of the preprocessing step (Section 2.3.1), DL models (Section 2.3.2, Section 2.3.3, Section 2.3.4, Section 2.3.5, Section 2.3.6 and Section 2.3.7), implementation details (Section 2.3.8) and applied XAI tools (Section 2.3.9) are presented.

#### 2.3.1. Preprocessing

Sensor data fed to the DL models were resampled to a common frequency of 256 Hz and min-max normalised per sample, as normalisation can have a significant impact on the classification accuracy [45]. Especially in Neural Networks (NNs), normalisation helps the network to extract meaningful features [46]. Moreover, the samples were smoothed using a moving-average algorithm with a window size of one second. The outcome of the smoothing step can be seen in Figure 5. No further preprocessing was performed, and DL architectures were trained directly on the time series data.

#### 2.3.2. Multi-Layer Perceptron

MLPs consist of interconnected artificial neurons grouped into so-called layers. Different layers are stacked to retrieve meaningful information from the given input data. Each neuron is linked to all preceding neurons of the previous layer, calculating its output as a weighted summation of its inputs plus a bias value. Usually, the output is also sent through a non-linear function referred to as an activation function to make the model able to learn non-linear relations between the data and labels. The simple calculation of one neuron can be processed using the following equation:(5)$$y=\sigma \left(b+\sum_{k=1}^{n}w_{k}x_{k}\right)\;$$
where σ in an activation function, *b* is a learnable bias value, $w_{k}$ is the learnable weights applied to the previous input $x_{k}$ and *n* is the number of neurons in the preceding layer. A deep MLP consists of at least 3 layers, i.e., one input, one hidden, and one output layer. Often, dropout layers are used to avoid overfitting the network by deactivating a proportion of random neurons during training [47]. For classification tasks, the number of neurons in the output layer is often set to the number of classes, and a result is presented using a softmax activation function, where the class output with the maximum value is interpreted as the predicted one. The MLP architecture used in this study is a simple feed-forward NN consisting of a flatten layer and 2 blocks with a dropout and a dense layer (250 and 100 neurons) with a Rectified Linear Unit (ReLU) activation function. The blocks are connected to a dense layer with 2 neurons and a softmax activation for classification.

#### 2.3.3. Convolutional Neural Network

In contrast to MLPs, CNNs have been designed to extract features from image data, and thus utilise convolutional kernels instead of individual artificial neurons. Here, the weights of convolutional kernels are learned and applied to the input of the layer to generate an output referred to as feature maps. To further improve the computation speed by reducing the input dimensionality, pooling layers that downsample their input are usually used as well. For example, max pooling layers aggregate the information of the previous layer by summarising areas of the input feature maps by their maximum value. Similarly to the convolutional layers with their kernel size, the pooling layers can also have their window (or pool) size adjusted. In the end, the features retrieved from the CNN architectures are flattened and fed to an MLP to obtain a classification output. The CNN model used in this study is inspired by the one in [20] and consists of 3 blocks of convolutional, max pooling and dropout layers.

#### 2.3.4. Convolutional Autoencoder

In addition to the end-to-end feature learning approaches of the MLP and CNN, a CAE was adopted for the task of automated pain recognition. The CAE aims to learn an embedding of the given data samples in an unsupervised way by reducing the dimensionality of the input (encoder) and reconstructing the input given just the embedding afterwards (decoder). Successful training of such an autoencoder results in an encoder that can retrieve important features from the input while the decoder is still able to reconstruct the input with minimal error. Then, a model can be trained for classification using the features computed by the encoder. To utilise the trained CAE model that is fitted in an unsupervised way on the sensor data using Mean Squared Error (MSE) as the loss between the input and reconstruction sample for a classification task, the encoder output (of the last max pooling layer) was flattened, processed by dense layers with 100, 50 and 25 neurons, and a classification outcome was computed by a dense layer with 2 neurons and a softmax activation function similarly to the other DL architectures. The new layers were then fine-tuned in a supervised manner to fit the given pain classification task.

#### 2.3.5. Supervised Contrastive Learning

Recent advances in DL have led to the introduction of a novel training procedure for NNs called supervised CL [41]. The two-step approach consists of a pre-training step where an encoder is trained to generate vector representations of the input data. The loss monitored during training of the encoder is the supervised contrastive loss that ensures that encodings of samples deriving from one class are alike, and encodings from varying classes are apart. Finally, the frozen encoder is connected to an MLP that is fine-tuned for classification. The encoder of the supervised CL approach was chosen to be similar to the presented one of the CAE (Table A2) with an additional projection head consisting of dense layers with 100, 50 and 25 neurons. The MLP for classification involves a dense layer with 100 neurons and a ReLU activation function and a dense layer with 2 neurons and a softmax activation function to present the output.

#### 2.3.6. Transformer Network

First introduced for natural language processing, the so-called transformers [42] make use of a multi-head self-attention mechanism. The self-attention mechanism accesses the trained knowledge of previous steps, and thus provides information about temporal aspects of the input data. Due to their architecture, which can handle sequential data, transformers can also be easily applied to time series data. Thus, the time series input is split into 8 parts of equal length.

#### 2.3.7. MDK-Resnet Architecture

To facilitate the ability of CNNs to process 2D data, a method called GAF [43] that transforms 1D time series into 2D representations was tested as well. The 1D time series data were encoded into images by polar coordinates-based matrices that can preserve absolute temporal correlation [48]. Therefore, the input data are normalised, the inverse cosine of the time series values is taken as the angle and the time label is taken as the radius in the polar coordinate system. Then, Gramian Summation Angular Field (GASF) and Gramian Difference Angular Field (GADF) matrices are computed by a trigonometric sum/difference between each point. Afterwards, the acquired image representations are fed to a MDK-Resnet architecture following the work of Xu et al. [44], which was evaluated on time series data for HAR in the past. It leverages MDK modules consisting of multiple different dilated kernels [49] applied to the input and merged by addition. Dilated kernels try to improve the receptive field of convolutional layers by artificially widening the kernels while maintaining the initial computational cost. An MDK module is composed of the following branches:Identity: The input is passed through.1 × 1: A 2D convolution with a kernel size of (1, 1).Dilation 1: A 2D convolution with a kernel size of (3, 3).Dilation 2: A 2D convolution with a kernel size of (3, 3) and a dilation rate set to (2, 2).Dilation 4: A 2D convolution with a kernel size of (3, 3) and a dilation rate set to (4, 4).

#### 2.3.8. Classification Step

To have a direct comparison between RF with HCFs and the DL models and to enable RFE for the latter, a combination of DL and RF models was implemented in addition to the end-to-end evaluation. It was realised by training DL models in an end-to-end manner, discarding the last classification layer and feeding the output as features to a RF model. The DL models were implemented using Keras 2.7 and trained with a batch size of 32, a learning rate set to $10^{-4}$ and on 100 epochs if not otherwise mentioned. EDA data samples were given to the models as a 3D shape in the form (time × sensor channels × 1) with one sensor channel (EDA) and the time steps according to the sampling rate and window size of the datasets. The resulting windows had the shape (2500 × 1 × 1) and (2816 × 1 × 1) for the PMDB and BVDB, respectively. To process the 3D data, 2D convolutions were used with kernel, pool and stride sizes equal to (*n*, 1), with *n* being the kernel size across the time dimension as defined in Table A1, Table A2, Table A3 and Table A4. As the only exception, the unsupervised pre-training step of the CAE was performed with a batch size of 8. Dropout layers were used in most architectures to avoid overfitting on the training sets. ML have been evaluated on a machine with an i7-7700K CPU, 16 RAM and Geforce GTX 1080 Ti.

#### 2.3.9. XAI in DL

Several attempts have been investigated in the past to increase not only the classification performance of DL models, but also their understandability. Because of its importance in ML, several techniques for image classification have been introduced to highlight areas of the input data that have a high impact on the classification outcome towards specific classes. Zhou et al. [50] introduced the so-called Class Activation Maps (CAMs) to calculate activation maps for CNN architectures involving a global average pooling layer followed by the softmax classification layer. Thus, the input importance is calculated by considering the outputs of these two layers that can be specified as follows. The global average pooling for unit *k* can be defined as:(6)$$F_{k}=\sum_{x,y}f_{k}\left(x,y\right),$$
where $f_{k}\left(x,y\right)$ represents the activation of unit *k* in the last convolutional layer. Moreover, the input to the softmax activation is given by:(7)$$S_{c}=\sum_{k}w_{k}^{c}F_{k},\;$$
where the learnable weight $w_{k}^{c}$ indicates the importance of $F_{k}$ for class *c*. By putting Equation (Equation 6) into Equation (Equation 7) and rearranging, the softmax input can be expressed as:(8)$$S_{c}=\sum_{k}w_{k}^{c}\sum_{x,y}f_{k}\left(x,y\right)=\sum_{x,y}\sum_{k}w_{k}^{c}f_{k}\left(x,y\right).\;$$

Then, the CAM for class *c* with each spatial element *x* and *y* is defined as:(9)$$M_{c}\left(x,y\right)=\sum_{k}w_{k}^{c}f_{k}\left(x,y\right).\;$$

To overcome the need for a specific architecture type, the Grad-CAM approach was introduced by Selvaraju et al. [51] in 2017. This generalisation of CAMs does not rely on architectures using a global average pooling layer, but applies to any CNN by using the gradient information flowing into the (last) convolutional layer of the CNN to understand the importance of each neuron for a decision of interest. Here, the importance weights are estimated by calculating the gradients of the backpropagation. First, the gradient of the score for class *c*, $y_{c}$ (before the softmax), with respect to feature map activations $f^{k}$ of a convolutional layer, i.e., $\frac{\partial y_{c}}{\partial f^{k}}$ [51], is computed. Moreover, global-average-pooling is applied to the width (*w*) and height (*h*) dimensions of the feature map to retrieve importance weights $a_{c}^{k}$ as follows:(10)$$a_{c}^{k}=\frac{1}{w+h}\sum_{i}^{w}\sum_{j}^{h}\frac{\partial y_{c}}{\partial f_{k}\left(i,j\right)}.\;$$

This weight captures the ‘importance’ of a given feature map *k* for a target class *c* and is thus summed as a weighted combination of all feature maps *K* to estimate an overall score $L_{c}$. In addition, a ReLU activation function is applied to focus on positive activation towards the given class:(11)$$L_{c}=ReLU\left(\sum_{k}^{K}a_{k}^{c}f^{k}\right).\;$$

## 3. Results

The following section gives an insight into the results that were achieved on both the PMDB and BVDB. The performances of the ML approaches presented in Section 2.2 and Section 2.3 are presented first. All reported classification performances are given as accuracy, as it is the standard metric for pain recognition [16,17,18,19,20] and provides a realistic representation of the results given the balanced classes of the pain recognition datasets. In order to give comparable results to previous publications (see Table 1) and test the algorithms in a subject-independent manner, all models were evaluated in a Leave-one-subject-out (LOSO) CV.

### 3.1. Comparison Study

Table 3 summarises the performance of the RF model based on HCFs and several end-to-end DL architectures trained for the task “no pain” vs. “high pain” on the BVDB and PMDB evaluated in a LOSO CV. The best results were yielded by the RF approach on the PMDB with an accuracy of 91.70%, whereas the best performance on the BVDB was obtained by the supervised CL approach with an accuracy of 84.54%.

Since the RF model performed the best and decent on the PMDB and BVDB, respectively, the RF model and its classification result were further analysed. Figure 6 shows a confusion matrix for the RF approach evaluated in a LOSO CV on both datasets. The reported numbers represent the cumulative results of each individual fold.

Moreover, Table 4 summarises the computational time for each model on the PMDB and BVDB in a Leave-one-subject-out Cross Validation. The results are reported as an average of five runs and given in seconds. While the LOSO is finished relatively quickly for the straightforward DL models such as MLP and CNN, the training time increases gradually for the transformer, MDK-Resnet, Supervised CL and CAE models. The RF training was the fastest on both datasets.

### 3.2. Interpretability Study

The results of the feature importance analysis of the RFs trained with HCF can be found in Table 5. The top 10 features with the most impact on classification performance evaluated in a LOSO manner were listed separately for the BVDB and PMDB. Although some complex features deriving from the dPhEDA approach are considered relevant for the target classification problem, the best features come mostly from elementary statistical calculations, such as “argmax” or the difference of the first and last value in the time series (“diff_start_end”). In addition, a significant gap between the importance of the first feature to the subsequent ones is existing in both datasets.

Figure 7 compares the LOSO accuracy outcome for several RFE tests for the different tested approaches on the BVDB and PMDB. In the RF approach, RFE was applied directly to the HCFs, whereas in the DL approaches, the sensor data were first transformed in a feature set using the trained NNs without the softmax classification layer. Afterwards, an RF with RFE was trained on the newly acquired feature set. The accuracies for both datasets are stable for most of the RFE iterations, but drop visibly for most models when 10 or fewer features remain. The best performance was obtained with 7 HCFs (“min”, “argmax”, “argmin”, “diff_start_end”, “mean_phasic”, “norm_var” and one feature from “dPhEDA”) on the BVDB achieving an accuracy of 83.74%, and with 19 HCFs (“max”, “min”, “iqr”, “rms”, “local_max”, “argmax”, “argmin”, “diff_start_end”, “sd_tonic”, “range_tonic”, seven features from “dPhEDA”, and two from “TVSymp”) on the PMDB with an accuracy of 93.26%. RFE leads to performance improvement in all scenarios, with a higher number of features required for optimal performance in DL models compared to RF. Table 6 summarises the accuracies without RFE, the number of optimal features and the best accuracies with RFE for the different approaches.

## 4. Discussion

The following section offers a detailed discussion concerning the presented results. First, the comprehensive investigations carried out with various ML approaches reveal that despite being stated differently in the past [13], HCFs can outperform the models based on automated feature learning methods. In particular, RFE can boost the performance of HCF-based approaches and broaden their superiority over DL techniques. One benefit of the leveraged RF model based on HCFs is the immediate exploitation of the raw data, whereas DL rely on a normalisation step to quicken convergence. Here, it is also assumed that the loss of information regarding the raw data values caused by the normalisation step explains the weaker performance results yielded by the NN architectures. Moreover, the training time of the RF is significantly shorter than the ones of the DL models. Nevertheless, all models yield comparable results as they managed to retrieve the needed information for the given task. Although the same approaches were applied on both BVDB and PMDB datasets, a difference of ∼10% in the best classification accuracy can be observed for “no pain” vs. “high pain” between the two datasets. This could be explained by the longer stimuli and time series windows in the PMDB that permit the detection of long responses in EDA. Despite intensive efforts to further optimise and improve the classification results of DL models, the performance yielded with HCFs still competes with DL. For this reason, the importance of single features for classification performance was investigated in the scope of pain. Results for our feature importance analysis (Table 5) suggest that classifiers trained on both datasets are dominated by individual characteristics rather than an accumulation of many features, clearly visible on the margin between the most dominant feature and consecutive ones. Moreover, RFE on both datasets indicates that 7 to 19 features are sufficient for the classification task of “no pain” vs. “high pain” based on EDA time series data. Here, a great proportion of features are basic statistical features describing the outer shape of the given data sample such as “min”, “max”, “argmin”, “argmax”, “diff_start_end”, “mean_phasic”, “norm_var”, “iqr”, “rms”, “sd_tonic”, and “range_tonic”.To further investigate the validity of simple statistical features, a small study in which simple naive features were calculated and used to individually classify “no pain” against “high pain” was conducted. These classification rules, based on simple Boolean tests, are carried out on a single feature of the EDA signal, either confirming the class “high pain” or rejecting a painful class, thus categorising the sample as “no pain”. The terms can be applied to all samples of a given dataset and be compared with the actual label to estimate the accuracy. Furthermore, the tested classification rules were found in a trial and error fashion, based on the analysis of the most important features derived from the feature importance analysis. For the given conditions, the EDA samples are considered as sequences expressed by $x_{i}$, with 1⩽i⩽l being the index of a value and *l* the length of the time series. First, the “diff_start_end” feature is evaluated on its own by checking whether the last element is greater than the first element of the given time series ($x_{0}<x_{l}$). Afterwards, calculations around the “argmin”/“argmax” in relation to the length of the time series data to evaluate when the highest values occur and how much the signal is increasing were tested. More specifically, it is tested whether the “argmax” is found after $\frac{7}{10}$ of the sample’s time and whether the difference between “argmax” and “argmin” is greater than $\frac{1}{4}$ of the time series length. Finally, categorisation is carried out by examining whether the signal tends to rise or fall by checking whether the sum of the discrete derivative is greater than zero. The discrete derivative of a time series *x* with length *l* is defined by
(12)$$x_{i}^{\prime }=x_{i}-x_{i-1}\;\forall \;2⩽i⩽l,$$
where $f_{1}^{\prime }$ is arbitrarily set to 0. This simplicity contrasts with the complexity of the training process of DL models and the complexity of the DL models themselves. Table 7 summarises the accuracies obtained with these classification rules based on simple features and compares them to the RF and MLP approaches on the PMDB and BVDB.

Similar to the RFE, various simplistic features could be introduced to classify EDA into “no pain” and “high pain” classes with comparable performance results to various ML models. It is important to note that no training of complex ML models is performed here, but the classification is based on a decision involving the computation of a single feature. It again underscores the point that the general shape of the time series and whether the EDA signal increases or decreases are important—even to the point that individual features are sufficient for classification. Different features showed varying results on both datasets, where some performed better on PMDB and some were preferable on the BVDB. Overall “argmax” and “argmin” values, which characterise the peak and low points of the EDA signal, and when they occur seem to be highly relevant for the given task. In addition, it should be noted that there could be even better features, as the ones presented (Table 7) were found through trial and error. The simple relationship between individual features and class association becomes apparent when the distribution of a single feature is plotted individually for the various classes. Figure 8 and Figure 9 show boxplots for the “argmin” feature for the classes of the PMDB and BVDB, respectively. The “argmin” value describes the position of the minimum element in the time series. After conversion into seconds, it describes at what point in time the minimum of the series is reached. Whereas high values are clearly dominant in low-pain classes, low values are outweighing those in high-pain classes. In simpler words, a decrease of EDA over time is often seen in resting phases without stimulus (thus minimum being at the end of the window samples), while an increase (and thus minimum at the beginning of window samples) is present in time series belonging to high pain classes. Again, the better classification results yielded on the PMDB outlined in greater differentiation of “no pain” vs. “high pain” in PMDB (*B* vs. $P_{4}$) in comparison to BVDB ($T_{0}$ vs. $T_{4}$) can be explained by the longer time windows in samples.

While retrieving the feature importance in classical ML models, as with the impurity score in RFs, is relatively simple, NNs lack deeper interpretation tools despite their recent success. Thus, the research community worked on the interpretability of such models by creating grey-box classifiers from black-box ones [52] or relying on white-box algorithms [53]. To not alter the classification system itself, the interpretability in NNs is mainly investigated in the form of heat maps that highlight areas in the input data that are relevant to the classification outcomes. Examples of the results of Grad-CAM on the PMDB for the CNN architecture can be seen in Figure 10 and Figure 11. The EDA data for the eight stimulus repetitions for the respective classes “no pain” (left) and “high pain” (right) of a subject are presented in two different graphs. In addition, Grad-CAM highlights once which parts are relevant for the class “no pain” (Figure 10) and once for the class “high pain” (Figure 10) with the colour coding specified by the existing colour bar on the right side. All of the subject’s samples were correctly classified by the CNN approach. See Section A.3 for more examples, especially for a subject where the DL architecture struggles to classify all the samples well.

The resulting plots differ greatly for the activation towards “no pain” and “high pain”. More specifically, areas with decreasing EDA are highlighted for the samples with no pain, while areas with increasing EDA are emphasised for the classification towards pain in contrast. Heat maps, similar to saliency, CAMs, Grad-CAM or grad-CAM++, have been investigated heavily on image databases in the past. Although the interpretation of these heat maps for images created by Grad-CAM is trivial, the output of these techniques is not clear for time series data, and an objective analysis remains challenging. From the examination of Figure 10 and Figure 11, it is apparent that there is a correlation between the slope of the signal and the class association of the CNN for the given subject. To evaluate this finding for all subjects of the dataset, a naive approach to objectify the class importance given by Grad-CAM with regard to the available classes and raising and falling slopes of the given time series data is presented. For this purpose, heat maps for all samples of the PMDB are calculated using Grad-CAM first. The CNN model used to retrieve a classification output and Grad-CAM heat map were trained in a LOSO manner as well. Next, the resulting maps are masked depending on the discrete derivative of the curve to estimate the focus of the NN for areas with positive and negative slopes in the samples with respect to the classification outcome for each class individually. An overall estimation is calculated by summing up and normalising all the masked scores of all available samples. In the following, the calculation of this score is described in more detail. To begin with, the following notation is used:Dataset: $D={\left\{x^{\left(i\right)}\right\}}_{i=1}^{n}$ where $x^{\left(i\right)}\in R^{l}$ and 1⩽i⩽n is the example index.Length of one data sample: *l*.Grad-CAM method: $L_{c}:R^{l}\to R^{l}$.

To investigate the activation during areas with positive and negative slopes of the time series, two helper functions $g^{+}\left(x\right)$ and $g^{-}\left(x\right)$ for x∈D to mask the activation are used. The function $g^{+}$ enables areas with a positive slope by leveraging only the outcome greater than 0 of the discrete derivative introduced in Equation (Equation 12) by: (13)$${\left[g^{+}\left(x\right)\right]}_{i}=\left\{\begin{array}{cc} 1 & \;\mathrm{if}x_{i}^{\prime }>0\\ 0 & \;\mathrm{otherwise}.\; \end{array}\right.$$

Afterwards, an importance score $w_{c,i}^{+}$ can be calculated for any given sample $x^{\left(i\right)}$ in relation to the class *c* and positive slope by masking the general Grad-CAM output with the created function $g^{+}$:(14)$$w_{c,i}^{+}=L_{c}\left(x^{\left(i\right)}\right)\cdot g^{+}\left(x^{\left(i\right)}\right)\;\forall \;1⩽i⩽n.$$

The above step is performed for all samples in the database. The individual results are summed up leading to a weights value, but representing the class activation for the given class in parts of a positive slope for the whole dataset:(15)$$w_{c}^{+}=\sum_{i=1}^{n}w_{c,i}^{+}.\;$$

By adopting Equation (Equation 13) to
(16)$${\left[g^{-}\left(x\right)\right]}_{i}=\left\{\begin{array}{cc} 1 & \;\mathrm{if}x_{i}^{\prime }⩽0\\ 0 & \;\mathrm{otherwise} \end{array}\right.$$
and updating Equations (Equation 14) and (Equation 15) accordingly, the activation for negative slope ($w_{c}^{-}$) can be calculated equally. To have an interpretable value between 0 and 1, and to compare it with other classes, the values are normalised per class at this point. Here, each weight value is divided by the sum of weights inside its class. The following equation summarises the normalisation step:(17)$$w_{c}^{+norm}=\frac{w_{c}^{+}}{w_{c}^{+}+w_{c}^{-}}$$
and
(18)$$w_{c}^{-norm}=1-w_{c}^{+norm}.$$

The resulting values for $w_{c}^{+norm}$ and $w_{c}^{-norm}$ represent a score estimating the importance of parts with increasing and decreasing EDA signals, respectively, for the classification outcome towards class *c*. Figure 12 visualises $w_{c}^{+norm}$ and $w_{c}^{-norm}$ for the classes “no pain” and “high pain” of the PMDB. For the class “high pain”, $w^{+}$ exceeds $w^{-}$, whereas $w^{-}$ outweighs $w^{+}$ for the class “no pain”. In other words, the attention scores, and thus the focus of the DL model, are mostly concentrated in areas where the EDA rises for “high pain”, and decreases for “no pain”.

Moreover, the presented importance score based on Grad-CAM for DL architectures indicates that simple characteristics of the time series samples have a high impact on the classification results similar to the HCF approach. In simple words, areas with positive and negative slopes highly influence the results towards the classes “high pain” and “no pain”, respectively. An SCR event is associated with pain, whereas no EDA reaction, i.e., constant or decreasing SCL, is associated with no pain. Despite their complexity and numerous parameters, NN architectures may learn simplistic characteristics of EDA samples in the scope of automated pain research. In summary, the in-depth analysis of the feature importance in RF models and the characteristics that are influencing the DL outcome shows that straightforward features are the most relevant for the task of automated pain recognition based on EDA data. The results showed that “argmin”, “argmax”, the difference between the first and last values, and the sum of the slope (derivative) of the given sequence are of high importance. Here, it is assumed that certain increases in EDA data, also referred to as SCR, are greatly connected to the classes associated with pain. Although not all samples with an SCR are categorised as painful, the two concepts are affiliated strongly. To further investigate possibilities to improve the classification performance, the wrongly classified samples for the best model (RF) on the PMDB were examined. For better readability, the samples were clustered using K-Means clustering based on DTW Barycenter Averaging and plotted for their respective classes individually. The number of clusters was found using the Elbow method. A more detailed description can be found in Section A.4. Figure 13 and Figure 14 summarise the samples for the classes “no pain” and “high pain”, respectively.

Samples that should be classified as “no pain” but were labelled as “high pain” (Figure 13) are clustered into four groups. Clusters 0, 2 and 3 seem to have a major increase in the EDA data, which could explain why the classifier decided to choose “high pain”. Probably, the SCR observed for these examples was triggered by another event not related to the study, as SCRs are not specific to pain but can be introduced by various incidents [54] (p. 13), such as strong emotions or a demanding task. The samples in cluster 1 seem to have little variation in the signal, so understanding why the classifier made the wrong decision is difficult. On the other hand, wrongly classified “high pain” samples are categorised into four clusters (Figure 14). The instances are clustered into groups with almost no variation (clusters 2 and 3) and small fluctuations (clusters 0 and 1), which are not unambiguously considered as pain by the ML models. The absence of strong reactions in the EDA signal can be reasoned by the existence of subjects with little EDA activity despite environmental factors, who are commonly referred to as non-responders in the literature [55]. Because of the aforementioned discrepancies between classes and association with autonomic responses in the EDA signal, it is believed that it will be difficult to further improve classification results for automated pain recognition on both the PMDB and BVDB significantly. This assumption is further underlined by the trend of smaller performance increments in novel proposed classification systems (as shown in Table 1), which suggests a “natural boundary of maximal classification performance” being present in datasets in relation to complex tasks such as pain recognition based on physiological sensor data.

## 5. Conclusions

In this paper, an in-depth comparison of various ML models based on feature engineering and end-to-end feature learning including recent state-of-the-art DL methods evaluated on the PMDB and BVDB was conducted. The best accuracy of 93.26% using an RF with RFE and 84.54% using a supervised CL approach could be achieved for the task “no pain” vs. “high pain” on the PMDB and BVDB, respectively. Moreover, different approaches for XAI in the scope of automated pain recognition based on EDA data were shown. The presented analysis showed that both classical ML approaches based on HCF, as well as DL models, rely on simplistic characteristics of the given time series data to classify pain. Rises in EDA (also referred to as SCR) are highly connected to pain, whereas no change or a slight decline is associated with the absence of pain. To summarise, our studies highlight the following insights: (1) single simplistic features can compete with complex DL models based on millions of parameters. (2) Both approaches, based on HCFs and DL features, focus on straightforward characteristics of the given time series data in the context of automated pain recognition. Moreover, the aforementioned points lead to the question of whether complex ML models such as NNs suit the task of heat-based automated pain recognition based on physiological signals. Although DL can obtain good performances, simplistic approaches based on simple features can achieve equal or better performances for a fraction of the cost (especially computational time). Therefore, instead of simply applying DL to any task, researchers should question its use for each situation individually. In the scope of automated pain recognition, straightforward features can yield a relevant classification performance while also giving insights into the decision-making process. Although novel insights for pain recognition using XAI were obtained, the presented approach of leveraging Grad-CAM for DL models is relatively naive. Future iterations of such methods should include further masking criteria than just the slope and gradient of the given time series samples. In particular, the mask presented in Equation (Equation 13) should be extended to grasp additional aspects of interest. For example, maximum and minimum in addition to further features for basic curve characterisation should be included to broaden XAI tools for deep learning applied to time series data and enhance the understanding of the role of EDA samples for automated pain recognition. Moreover, transfer learning approaches between pain recognition datasets could be investigated to check if it is possible to leverage publicly available data to improve the classification performances on a target dataset.

## Figures and Tables

**Figure 1 sensors-23-01959-f001:**
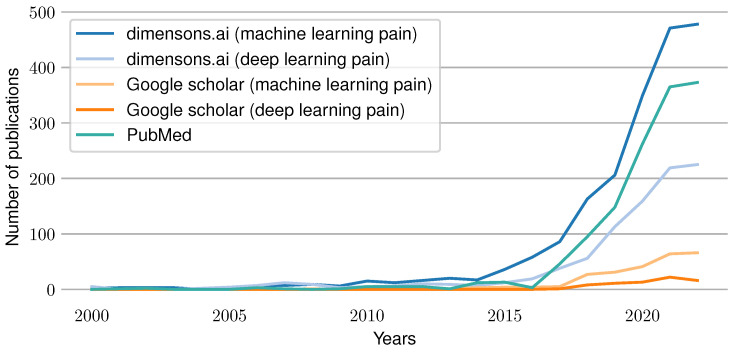
Publications per year in the field of pain prediction using machine learning techniques. Results are visualized for the following search engines and terms: dimensions.ai: “machine learning pain” in the title and abstract, dimensions.ai: “deep learning pain” in the title and abstract, PubMed: “pain machine learning[Title/Abstract]”, Google Scholar: “machine learning pain” in the title and Google Scholar: “neural network pain” in the title.

**Figure 2 sensors-23-01959-f002:**
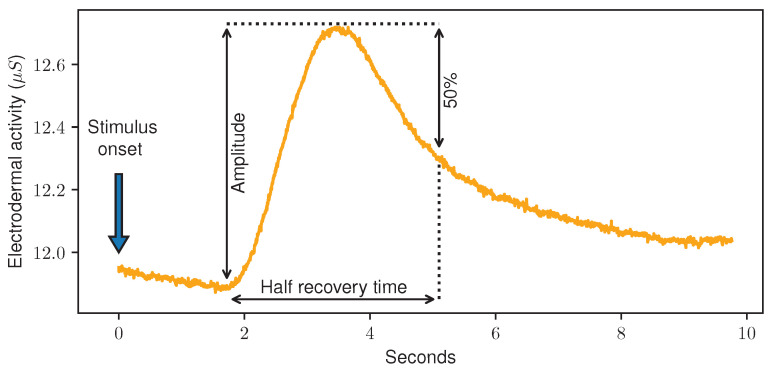
Electrodermal Activity data sample (orange) during a painful stimulus (blue) of the PainMonit Database. In addition, the amplitude of and half recovery time of the Skin Conductance Response is shown (black).

**Figure 3 sensors-23-01959-f003:**
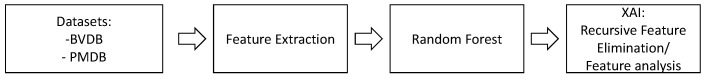
The flow of the applied feature engineering techniques. Initially, features are computed for the raw data samples of the PainMonit Database (PMDB) and BioVid Heat Pain Database (BVDB) datasets. Next, Random Forest (RF) models are trained for automated pain classification. Finally, the decision process is analysed using techniques of Explainable Artificial Intelligence, e.g., Recursive Feature Elimination (RFE).

**Figure 4 sensors-23-01959-f004:**
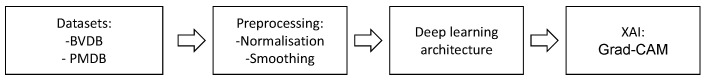
The flow of the applied feature learning techniques. Initially, the data samples of the PainMonit Database (PMDB) and BioVid Heat Pain Database (BVDB) datasets are normalised and smoothed. Next, several DL architectures are trained in an end-to-end manner. Finally, the decision process is analysed using techniques of Explainable Artificial Intelligence such as Gradient-weighted Class Activation Mapping (Grad-CAM).

**Figure 5 sensors-23-01959-f005:**
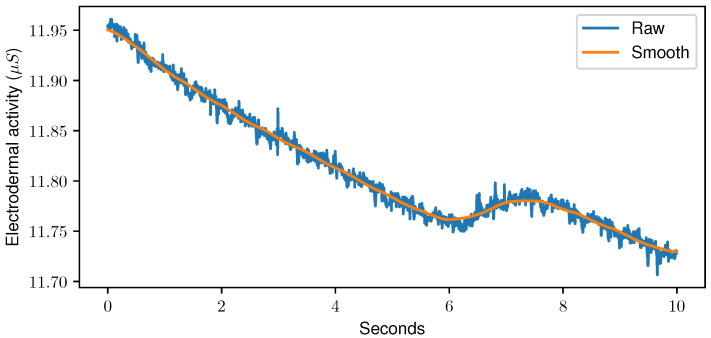
Raw data (blue) and outcome of the moving-average algorithm with a window size set to one second (orange) for a random sample of the Electrodermal Activity sensor of the PainMonit Database.

**Figure 6 sensors-23-01959-f006:**
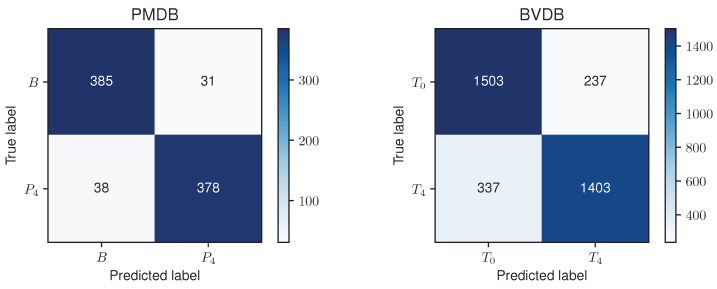
Confusion matrices for the Random Forest approach evaluated on the PainMonit Database (PMDB) (**left**) and BioVid Heat Pain Database (BVDB) (**right**) in a Leave-one-subject-out Cross Validation. An accuracy of 91.69% and 83.51% were obtained on the PMDB and BVDB, respectively.

**Figure 7 sensors-23-01959-f007:**
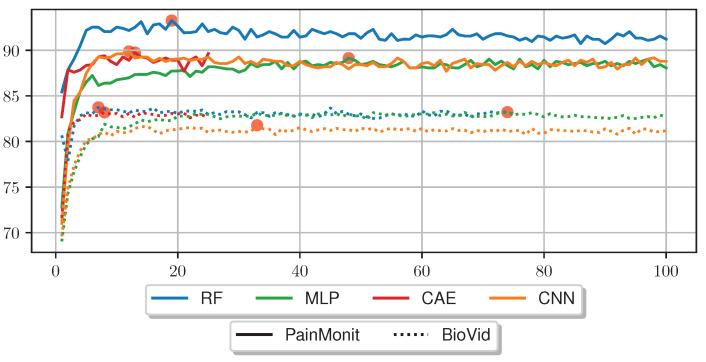
Comparison of different Recursive Feature Elimination parameters (x-axis) and the resulting accuracy of an Random Forest evaluated on the PainMonit Database and BioVid Heat Pain Database in a Leave-one-subject-out Cross Validation. Features were limited to a total of 100 for this plot.

**Figure 8 sensors-23-01959-f008:**
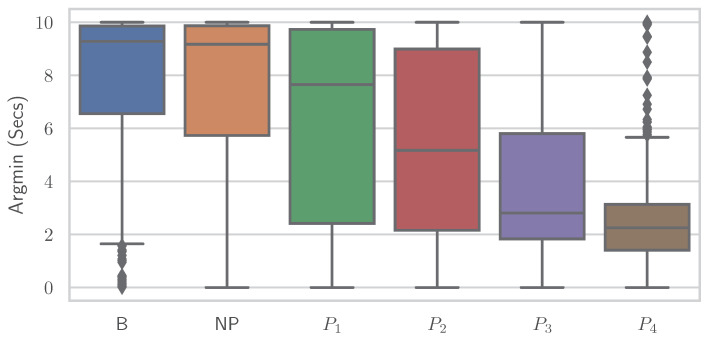
The “argmin” values of all Electrodermal Activity samples of the PainMonit Database visualised as boxplots for the different classes.

**Figure 9 sensors-23-01959-f009:**
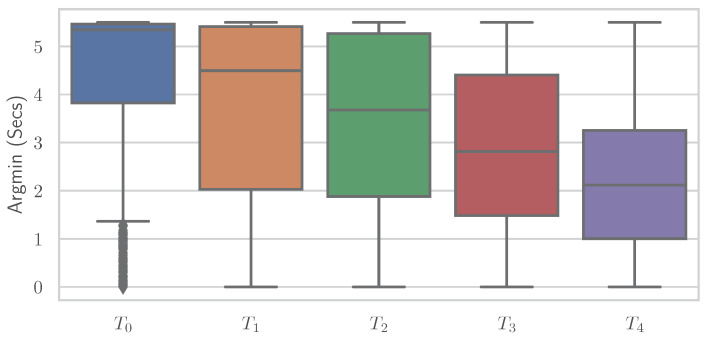
The “argmin” values of all Electrodermal Activity samples of the BioVid Heat Pain Database visualised as boxplots for the different classes.

**Figure 10 sensors-23-01959-f010:**
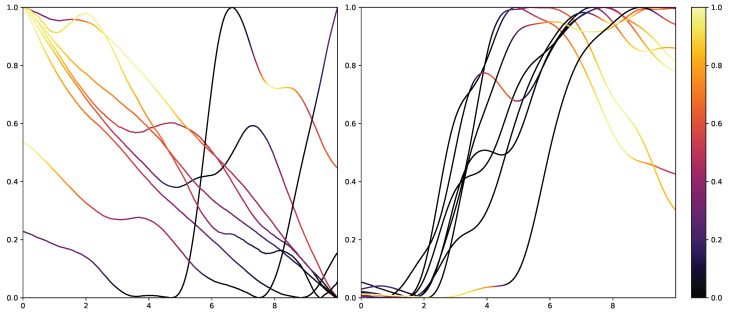
Electrodermal Activity samples for the class “no pain” (**left**) and “high pain” (**right**) for one subject with class activation calculated using Grad-CAM towards the class “no pain”. The time series data were smoothed by applying a moving average with a window size of one second.

**Figure 11 sensors-23-01959-f011:**
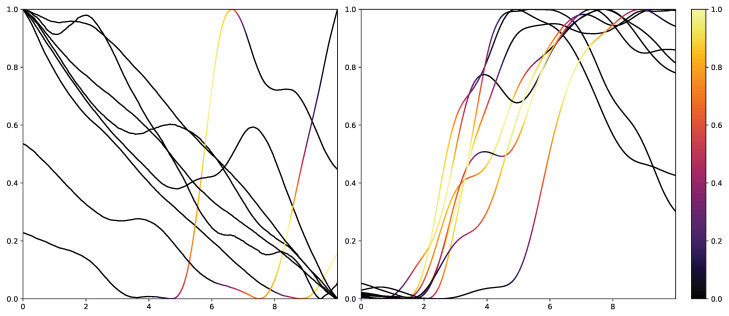
Electrodermal Activity samples for the class “no pain” (**left**) and “high pain” (**right**) for one subject with class activation calculated using Grad-CAM towards the class “high pain”. The time series data were smoothed by applying a moving average with a window size of one second.

**Figure 12 sensors-23-01959-f012:**
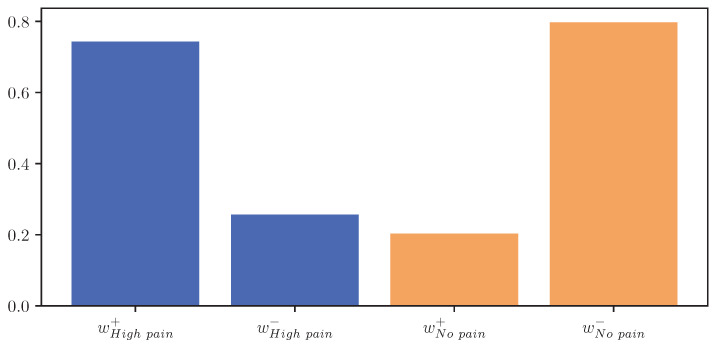
Important scores calculated for areas with positive (+) and negative (−) slope for the classes “no pain” (orange) and “high pain” (blue) of the PainMonit Database for the Convolutional Neural Network architecture visualised as a bar chart.

**Figure 13 sensors-23-01959-f013:**
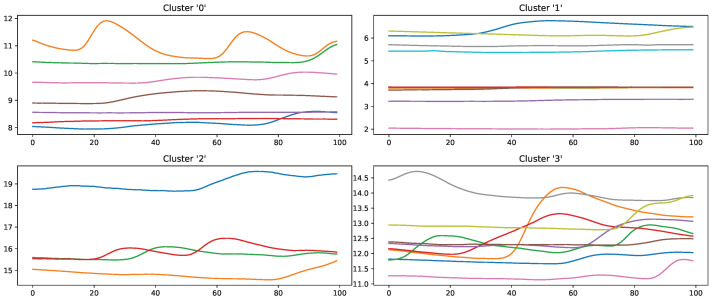
Samples of the class “no pain” wrongly classified by an Random Forest classifier in a Leave-one-subject-out evaluation scheme on the PainMonit Database, clustered into 4 groups using K-Means clustering based on DTW Barycenter Averaging.

**Figure 14 sensors-23-01959-f014:**
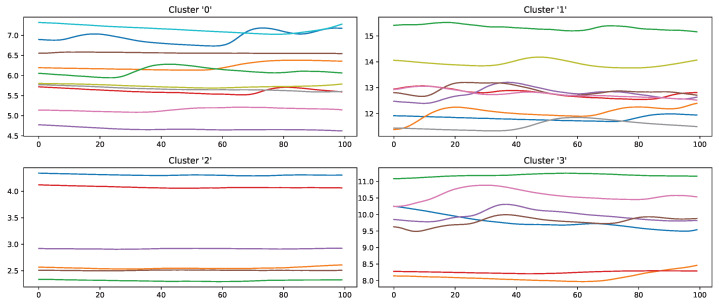
Samples of the class “high pain” wrongly classified by an Random Forest classifier in a Leave-one-subject-out evaluation scheme on the PainMonit Database, clustered into 4 groups using K-Means clustering based on DTW Barycenter Averaging.

**Table 1 sensors-23-01959-t001:** Leave-one-subject-out accuracy performance comparison of earlier work on the BioVid Heat Pain Database for the task $T_{0}$ vs. $T_{4}$.

Author	Year	Method	Sensors	$T_{0}$ vs. $T_{4}$
Werner et al. [16]	2014	Early Fusion with Random Forests	EMG, ECG, EDA	74.10
Kächele et al. [17]	2017	Early Fusion with Random Forests	EMG, ECG, EDA	82.73
Lopez-Martinez and Picard [18]	2018	Logistic Regression	EDA	74.21±17.54
Thiam et al. [13]	2019	Convolutional Neural Network	EDA	84.57±14.13
Thiam et al. [19]	2020	Gated Latent Representation	EMG, ECG, EDA	83.99±15.58
Gouverneur et al. [20]	2021	Multi-Layer Perceptron	EDA	84.22±13.86

**Table 2 sensors-23-01959-t002:** Hand-Crafted Features computed for the Electrodermal Activity signal.

Features
root mean square (RMS)
mean value of local maxima & minima
mean absolute value
mean of the absolute values (mav) of the first differences (mavfd)
mavfd on standardised signal
mav of the second differences (mavsd)
mavsd on standardised signal
variation of the first and second moment
indices of the minimum & maximum values
difference first and last value
mean & SD for phasic, tonic, amplitudes, rise times half recovery and recovery
range of tonic; number of SCRs
sum of amplitudes; first amplitude; phasic max;
mean, SD and Variance (VAR) on normalised signal

**Table 3 sensors-23-01959-t003:** Accuracy scores of several classifiers trained for the task “no pain” vs. “high pain” on the BioVid Heat Pain Database (BVDB) and PainMonit Database (PMDB) evaluated in a Leave-one-subject-out Cross Validation. The results of the best-performing method are depicted in bold for each dataset.

Approach	PMDB	BVDB
RF	**91.70 ± 7.83**	83.56 ± 14.94
MLP	91.09±9.70	84.22 ± 13.19
CNN	91.05±10.03	82.16 ± 14.08
CAE	90.58 ± 10.19	84.02 ± 13.09
Supervised CL	90.47 ± 10.37	**84.54 ± 12.82**
Transformer	90.61 ± 7.94	84.34 ± 12.60
MDK-Resnet	89.79 ± 11.68	81.50 ± 13.88

**Table 4 sensors-23-01959-t004:** Computation time (secs) of several classifiers trained for the task “no pain” vs. “high pain” on the BioVid Heat Pain Database (BVDB) and PainMonit Database (PMDB) evaluated in a Leave-one-subject-out Cross Validation.

Approach	PMDB	BVDB
RF	138	933
MLP	442	1074
CNN	389	3682
CAE	2906	18,346
Supervised CL	2216	12,164
Transformer	1673	10,207
MDK-Resnet	2180	14,197

**Table 5 sensors-23-01959-t005:** The 10 most discriminating features for the Electrodermal Activity signal calculated using the impurity scores of the Random Forest models in a Leave-one-subject-out Cross Validation. Scores were computed for the PainMonit Database (PMDB) and BioVid Heat Pain Database (BVDB) datasets. The importance scores were averaged across folds, min-max-normalised between 0 and 1 and rounded to the third decimal.

Feature	PMDB		BVDB	
Rank	Feature Name	Importance	Feature Name	Importance
10	dPhEDA_16	0.015	norm_sd	0.031
9	dPhEDA_4	0.016	norm_var	0.032
8	range_tonic	0.016	local_min	0.037
7	dPhEDA_6	0.016	dPhEDA_4	0.037
6	argmax	0.020	local_max	0.039
5	iqr	0.022	norm_mean	0.045
4	sd_tonic	0.028	mean_phasic	0.062
3	dPhEDA_3	0.030	diff_start_end	0.063
2	argmin	0.095	argmin	0.169
1	diff_start_end	1	argmax	1

**Table 6 sensors-23-01959-t006:** Classification results achieved by the tested models in combination with a Random Forest trained on the Electrodermal Activity signal in a Leave-one-subject-out with (column “Acc w/. RFE”) and without Recursive Feature Elimination (column “Acc w/o. RFE”). In addition, the optimal number of selected features is shown in the column “# RFE”.

Model	PMDB			BVDB		
	Acc w/o. RFE	# RFE	Acc w/. RFE	Acc w/o. RFE	# RFE	Acc w/. RFE
RF	91.22	19	93.26	82.96	7	83.74
MLP	88.07	48	89.14	82.67	74	83.22
CNN	88.79	12	89.87	81.18	33	81.78
CAE	89.64	13	89.75	82.61	8	83.19

**Table 7 sensors-23-01959-t007:** Overview of the accuracies obtained by simple classification rules in comparison to Machine Learning models for the classification of “no pain” vs. “high pain” on the PainMonit Database (PMDB) and BioVid Heat Pain Database (BVDB). The Electrodermal Activity samples are expressed by $\langle x_{i}|i\in I\rangle$, where *I* is the index of each element and *l* is the length.

Classification Rule	PMDB	BVDB
$x_{0}<x_{l}$	88.46	81.78
$\left(\frac{7l}{10}\right)<argmax_{i}\left(x_{i}\right)$	84.98	82.87
$\frac{l}{4}<\left(argmax_{i}\left(x_{i}\right)-argmin_{i}\left(x_{i}\right)\right)$	90.14	81.03
$0<\sum_{i=1}^{l}x_{i}^{\prime }$	88.46	81.78
RF	91.70	83.56
MLP	91.09	84.22

## Data Availability

The BVDB is available at http://www.iikt.ovgu.de/BioVid.html (accessed on 6 February 2023). The code is publicly available at https://github.com/gouverneurp/XAIinPainResearch (created on 27 January 2023). Further data sharing is not applicable to this article.

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
