# Peer review of "Explainable Artificial Intelligence (XAI) in Pain Research: Understanding the Role of Electrodermal Activity for Automated Pain Recognition"

_sensors, 2023, doi:10.3390/s23041959_

Round 1
Reviewer 1 Report
Check that the style of writing is in the third person throughout. Don’t use ‘we’.
Check that the abstract provides an accurate synopsis of the paper. It is very vague in its present form.
CNN architectures can be moved to annexures.
Was the algorithm trained using standard hyperparameters, or were they altered?
Comment on computational time and complexity in the training of the algorithm.
The authors do not mention the availability of this platform or framework for other practitioners, which is mandatory for this type of application.
You can refer following articles to lure potential readers.
A White-Box SVM Framework and its Swarm-Based Optimization for Supervision of Toothed Milling Cutter through Characterization of Spindle Vibrations
A Grey-Box Ensemble Model Exploiting Black-Box Accuracy and White-Box Intrinsic Interpretability
SD and Variance are essentially the same. Did you use both? Why? Justify.
There should be at least one confusion matrix in the results section to see the correct classification and misclassification.
How to deal with the data diversity of the present moment and moment in the future?
Hyperparameters of random forest tree must be included. Discuss the effect of n-estimators on the performance of tree. Which criterion was used while training random forest? Was it Gini or Entropy? Why? Did you check the influence of ‘max_depth’ and ‘min_samples_split’ on the testing of the tree? If not, this study is a must. You may refer to the recent article ‘Augmentation of Decision Tree Model Through Hyper-Parameters Tuning for Monitoring of Cutting Tool Faults Based on Vibration Signatures’
Author Response
Thank you for your valuable feedback! We tried to address specific requests. Answers to your comments can be found in the attached PDF.

Reviewer 2 Report
General Comments:
1) The content is acceptable and conforms to the scope of Sensors. However, improvements on the presentation of content is required.
2) Authors have presented on the significance of using simplistic manually crafted features and machine learning for pain recognition.
3) The use of straight forward characteristics of given information is vital for traditional feature engineering and deep learning approaches.
4) However, it is difficult to establish the necessity for an automated pain recognition system. What are its potential uses?
5) A total of forty-nine reliable sources have been cited throughout the manuscript.
Specific Comments:
1) The flow of methodology is not clear. Authors should include a flowchart of framework that indicates the sequential flow of experiments that contributes to the outcome of the study.
2) Generally, the role of electrodermal activity should be investigated first. After dwelling on the physiological aspects and the characteristic information, only then can the authors discuss on deep learning approaches.
3) The selection of machine learning and deep learning methods are not well-justified. Six deep learning methods have been adopted. The reason for selecting each of them are not clearly stated.
4) How are the feature information used with each of the deep learning methods? Specifically, how are the features presented? Are they in time-series format?
5) CNN-based methods are used for 2D information. Are there specific tests conducted to validate the best representation of the 2D information? Are the same format used for all the deep learning methods?
6) Every deep learning method can be optimized. The optimization process involves running specific experiments on different hyperparameters and tune them for optimum performance. Have these been considered in your study?
7) As the electrodermal activity are presented as time-series information, why haven't the authors adopt approaches such as long short-term memory neural network?
8) Elaboration of the methods are too shallow. A more in-depth explanation on each of the deep learning methods used must be discussed. Include equations that govern the techniques.
9) Deep learning methods require dedicated datasets for training and testing. What are the training-to-testing split ratio being used? What measures are being used to avoid overfitting?
10) What are the performance metrics used? Please mention explicitly in the manuscript. Include related equations and justify the use of those parameters.
11) Why didn't the study adopt methods such as principal component analysis to rank the features? What are the advantages of using the impurity score?
Author Response

(The authors gave the same response as above.)

Reviewer 3 Report
Overall, the manuscript is interesting, comprehensive, and very well-written.
1. Section 2.3.1 seems out of place (2.3.1 but no 2.3.2)
2. Section 2.4.2: why was this CNN specifically chosen?
3. Was transfer learning used in any of the experiments?
4. L273: 1-D?
5. L378: argmax as in Table?
6. Table headers should be above the tables
Author Response

(The authors gave the same response as above.)

Round 2
Reviewer 1 Report
Authors have addressed all my comments positively